# Transmission of SARS-CoV-2 in domestic cats imposes a narrow bottleneck

**Katarina M. Braun**[1], **Gage K. Moreno**[2], **Peter J. Halfmann**[1,3], **Emma B. Hodcroft**[4], **David A. Baker**[2], **Emma C. Boehm**[1], **Andrea M. Weiler**[1,5], **Amelia K. Haj**[2], **Masato Hatta**[1,3], **Shiho Chiba**[1,3], **Tadashi Maemura**[1,3], **Yoshihiro Kawaoka**[1,3], **Katia Koelle**[6], **David H. O'Connor**[2,5], **Thomas C. Friedrich**[1,5]*

1 Department of Pathobiological Sciences, University of Wisconsin-Madison, Madison, Wisconsin, United States of America, 2 Department of Pathology and Laboratory Medicine, University of Wisconsin-Madison, Madison, Wisconsin, United States of America, 3 Influenza Research Institute, School of Veterinary Sciences, University of Wisconsin-Madison, Madison, Wisconsin, United States of America, 4 Institute of Social and Preventative Medicine, University of Bern, Bern, Switzerland, 5 Wisconsin National Primate Research Center, University of Wisconsin-Madison, Madison, Wisconsin, United States of America, 6 Department of Biology, Emory University, Atlanta, Georgia, United States of America

☯ These authors contributed equally to this work.

* tfriedri@wisc.edu

**Data Availability Statement:** Source data after mapping have been deposited in the Sequence Read Archive (SRA) under bioproject

## Abstract

The evolutionary mechanisms by which SARS-CoV-2 viruses adapt to mammalian hosts and, potentially, undergo antigenic evolution depend on the ways genetic variation is generated and selected within and between individual hosts. Using domestic cats as a model, we show that SARS-CoV-2 consensus sequences remain largely unchanged over time within hosts, while dynamic sub-consensus diversity reveals processes of genetic drift and weak purifying selection. We further identify a notable variant at amino acid position 655 in Spike (H655Y), which was previously shown to confer escape from human monoclonal antibodies. This variant arises rapidly and persists at intermediate frequencies in index cats. It also becomes fixed following transmission in two of three pairs. These dynamics suggest this site may be under positive selection in this system and illustrate how a variant can quickly arise and become fixed in parallel across multiple transmission pairs. Transmission of SARS-CoV-2 in cats involved a narrow bottleneck, with new infections founded by fewer than ten viruses. In RNA virus evolution, stochastic processes like narrow transmission bottlenecks and genetic drift typically act to constrain the overall pace of adaptive evolution. Our data suggest that here, positive selection in index cats followed by a narrow transmission bottleneck may have instead accelerated the fixation of S H655Y, a potentially beneficial SARS-CoV-2 variant. Overall, our study suggests species- and context-specific adaptations are likely to continue to emerge. This underscores the importance of continued genomic surveillance for new SARS-CoV-2 variants as well as heightened scrutiny for signatures of SARS-CoV-2 positive selection in humans and mammalian model systems.

PRJNA666926. Derived data, analysis pipelines, and figures have been made available for easy replication of these results at a publicly-accessible GitHub repository: https://github.com/katarinabraun/SARSCoV2_transmission_in_domestic_cats.

**Funding:** This project was funded in part through a COVID-19 Response grant from the Wisconsin Partnership Program at the University of Wisconsin School of Medicine and Public Health to TCF and DHO. Author GKM is supported by an NLM training grant to the Computation and Informatics in Biology and Medicine Training Program (NLM 5T15LM007359). The funders had no role in study design, data collection, and analysis, decision to publish, or preparation of the manuscript.

**Competing interests:** The authors have declared that no competing interests exist.

## Author summary

Through ongoing human adaptation, spill-back events from other animal intermediates, or with the distribution of vaccines and therapeutics, the landscape of SARS-CoV-2 genetic variation is certain to change. The evolutionary mechanisms by which SARS-CoV-2 will continue to adapt to mammalian hosts depend on genetic variation generated within and between hosts. Here, using domestic cats as a model, we show that within-host SARS-CoV-2 genetic variation is predominantly influenced by genetic drift and purifying selection. Transmission of SARS-CoV-2 between hosts is defined by a narrow transmission bottleneck, involving 2–5 viruses. We further identify a notable variant at amino acid position 655 in Spike (H655Y), which arises rapidly and is transmitted in cats. Spike H655Y has been previously shown to confer escape from human monoclonal antibodies and is currently found in over 1,000 human sequences. Overall, our study suggests species- and context-specific adaptations are likely to continue to emerge, underscoring the importance of continued genomic surveillance in humans and non-human mammalian hosts.

## Introduction

Understanding the forces that shape genetic diversity of RNA viruses as they replicate within, and are transmitted between, hosts may aid in forecasting the future evolutionary trajectories of viruses on larger scales. The level and duration of protection provided by vaccines, therapeutics, and natural immunity against severe acute respiratory syndrome coronavirus 2 (SARS-CoV-2) will depend in part on the amount of circulating viral variation and the rate at which adaptive mutations arise within hosts, are transmitted between hosts, and become widespread. Here, to model the evolutionary capacity of SARS-CoV-2 within and between hosts, we characterize viral genetic diversity arising, persisting, and being transmitted in domestic cats.

A translational animal model can serve as a critical tool to study within- and between-host genetic variation of SARS-CoV-2 viruses. SARS-CoV-2 productively infects Syrian hamsters, rhesus macaques, cynomolgus macaques, ferrets, cats, and dogs in laboratory experiments. Natural infection with SARS-CoV-2 has also been documented in ferrets, mink, dogs, and small and large cats. This makes each of these potentially viable animal models, apart from large cats which are not typically used in biomedical research [1–5]. Among these species, natural transmission has only been observed in mink, cats, and ferrets [1,6,7]. Transmission from humans to mink and back to humans has also been recently documented [8]. Infectious virus has been recovered from various upper- and mid-respiratory tissues in cats and ferrets, including nasal turbinates, soft palate, tonsils, and trachea [1,6]. However, only in cats has infectious virus been recovered from lung parenchyma, where infection is most commonly linked to severe disease in humans [1,6,9,10].

Transmission bottlenecks, dramatic reductions in viral population size at the time of transmission, play an essential role in the overall pace of respiratory virus evolution [11–20]. For example, in humans airborne transmission of seasonal influenza viruses appears to involve a narrow transmission bottleneck, with new infections founded by as few as 1–2 genetically distinct viruses [12,13,16–18]. In the absence of selection acting during a transmission event, the likelihood of a variant being transmitted is equal to its frequency in the index host at the time of transmission (e.g. a variant at 5% frequency, has a 5% chance of being transmitted) [21]. When transmission involves the transfer of very few variants and selection is negligible, even beneficial variants present at low frequencies in the transmitting host are likely to be lost. Accordingly, although antigenic escape variants can sometimes be detected at very low levels

in individual human hosts, transmission of these variants has not been observed in nature [22–23]. In this way, narrow transmission bottlenecks are generally expected to slow the pace of seasonal influenza virus adaptation [11,24] and may have similar effects on SARS-CoV-2.

Accurate estimates of the SARS-CoV-2 transmission bottleneck size will therefore aid in forecasting future viral evolution. Previous studies have reported discordant estimates of SARS-CoV-2 transmission bottleneck sizes in humans, ranging from "narrow" bottlenecks involving 1–8 virions to "wide" bottlenecks involving 100–1,000 virions [25–28]. However, studies of natural viral transmission in humans can be confounded by uncertainties regarding the timing of infection and directionality of transmission, and longitudinal samples that can help resolve such ambiguities are rarely available. Animal models overcome many of these uncertainties by providing access to longitudinal samples in well-defined index and contact infections with known timing.

Here we use a cat transmission model to show that SARS-CoV-2 genetic diversity is largely shaped by genetic drift and purifying selection, with the notable exception of a single variant in Spike at residue 655 (H655Y). These findings are in broad agreement with recent analyses of evolutionary forces acting on SARS-CoV-2 in humans, suggesting human SARS-CoV-2 isolates are relatively well-adapted to feline hosts [25–32]. While estimates of the size of the SARS-CoV-2 transmission bottleneck remain highly discordant in humans, we find very narrow transmission bottlenecks in cats, involving transmission of only 2–5 viruses. Our findings show cat models recapitulate key aspects of SARS-CoV-2 evolution in humans and we posit that the cat transmission model will be useful for investigating within- and between-host evolution of SARS-CoV-2 viruses.

## Results

### Within-host diversity of SARS-CoV-2 in cats is limited

Recently, members of our team inoculated three domestic specific-pathogen free cats with a second-passage SARS-CoV-2 human isolate from Tokyo (hCoV-19/Japan/UT-NCGM02/2020) [33]. Each index cat was co-housed with a contact cat beginning on day 1 post-inoculation (DPI). No new cat infections were performed for this study. Nasal swabs were collected daily up to 10 days post-inoculation, **Fig 1**. Viral RNA burden is plotted in **S1A Fig** and infectious viral titers are shown in **S1B Fig**.

Using conservative frequency thresholds previously established for tiled-amplicon sequencing, we called within-host variants (both intrahost single-nucleotide variants "iSNVs" and short insertions and deletions "indels") throughout the genome against the inoculum SARS-CoV-2 reference (Genbank: MW219695.1) [34,35]. Variants were required to be present in technical replicates at ≥3% and ≤97% of sequencing reads [36] (all within-host variants detected at >97% frequency were assumed to be fixed; see Methods for details). iSNVs were detected at least once at 38 different genome sites. Of the 38 unique variants, 14 are synonymous changes, 23 are nonsynonymous changes, and one occurs in an intergenic region; this distribution is broadly similar to recent reports of SARS-CoV-2 variation in infected humans [30]. Similarly, we detected indels occurring at 11 different genome sites across all animals and timepoints. We identified 6–19 distinct variants per cat, of which 4–7 were observed on two or more days over the course of the infection within each cat (**S2 Fig**). All variants (iSNVs and indels) are plotted by genome location and frequency in **Fig 2A**.

### Genetic drift and purifying selection shape within-host diversity

To probe the evolutionary pressures shaping SARS-CoV-2 viruses within hosts, we first evaluated the proportion of variants shared between cats. Eighty-nine percent of variants (34 of 38

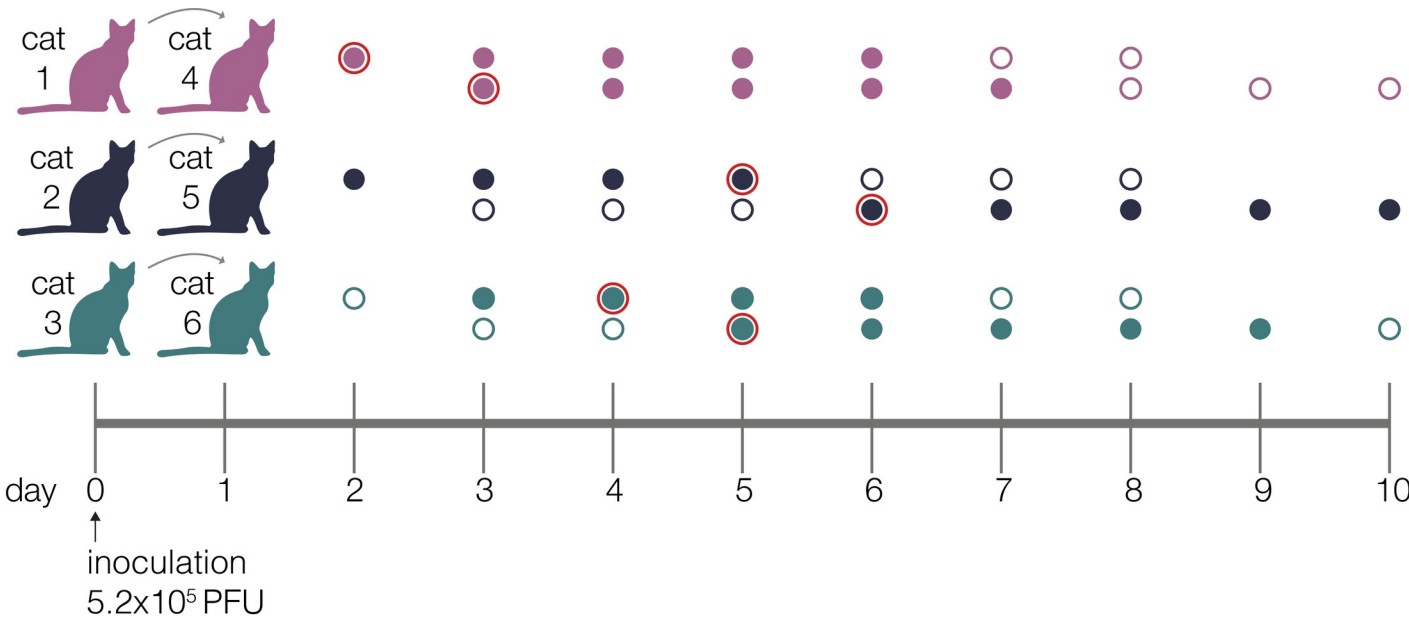

**Fig 1. Experimental timeline.** Schematic representing the sampling timeline for the three transmission pairs. Index cats were inoculated on day 0 with 5.2e5 PFU of a human isolate (hCoV-19/Japan/UT-NCGM02/2020) and were co-housed with a naive cat starting on day 1. Within each transmission pair, the top row of circles represents the index cat and the bottom row represents the contact cat. Open circles represent days on which there was no detectable infectious virus as indicated by plaque assay, and closed circles highlight days when live virus was recovered. Circles with a red outline indicate timepoints which were used in the beta-binomial estimate to calculate transmission bottleneck sizes.

iSNVs and 8 of 11 indels) were found in a single cat (42/49), 8% of variants were found in 2–5 cats (4/49), and the remaining 6% of variants were found in all 6 cats (3/49).

Purifying selection, which acts to purge deleterious mutations from a population, is known to result in an excess of low-frequency variants. In contrast, positive selection results in the accumulation of intermediate- and high-frequency variation [37]. Especially in the setting of an acute viral infection, exponential population growth is also expected to result in an excess of low-frequency variants [38]. To determine the type of evolutionary pressure acting on SARS-CoV-2 in cats, we plotted these distributions against a simple "neutral model" (light grey bars in **Fig 2B**), which assumes a constant population size and the absence of selection [37]. This model predicted that ~43% of polymorphisms would fall in the 3–10% frequency bin, ~25% into the 10–20% bin, ~14% into the 20–30% bin, ~10% into the 30–40% bin, and ~8% into the 40–50% bin. The frequency distribution of variants detected in each index cat across all available timepoints did not differ significantly from this "neutral" expectation ($p = 0.265$, $p = 0.052$, $p = 0.160$, respectively; Mann Whitney U test).

Next we compared nonsynonymous ($\pi N$) and synonymous ($\pi S$) pairwise nucleotide diversity to further evaluate the evolutionary forces shaping viral populations in index and contact animals [39]. Broadly speaking, excess nonsynonymous polymorphism ($\pi N/\pi S > 1$) points toward diversifying or positive selection while excess synonymous polymorphism ($\pi N/\pi S < 1$) indicates purifying selection. When $\pi N / \pi S$ is approximately 1, genetic drift, i.e., stochastic changes in the frequency of viral genotypes over time, can be an important force shaping genetic diversity. We observe that $\pi S$ exceeds or is approximately equal to $\pi N$ in most genes, although there is substantial variation among genes and cats (**S1 Table, S10 and S11 Figs**). $\pi S$ is significantly higher than $\pi N$ in all 3 index cats in Spike ($p = 0.005$, $p = 0.004$, $p = 0.019$, unpaired t-test) and ORF1ab ($p = 2.11e-05$, $p = 1.84e-06$, $p = 1.99e-06$, unpaired t-test) and in

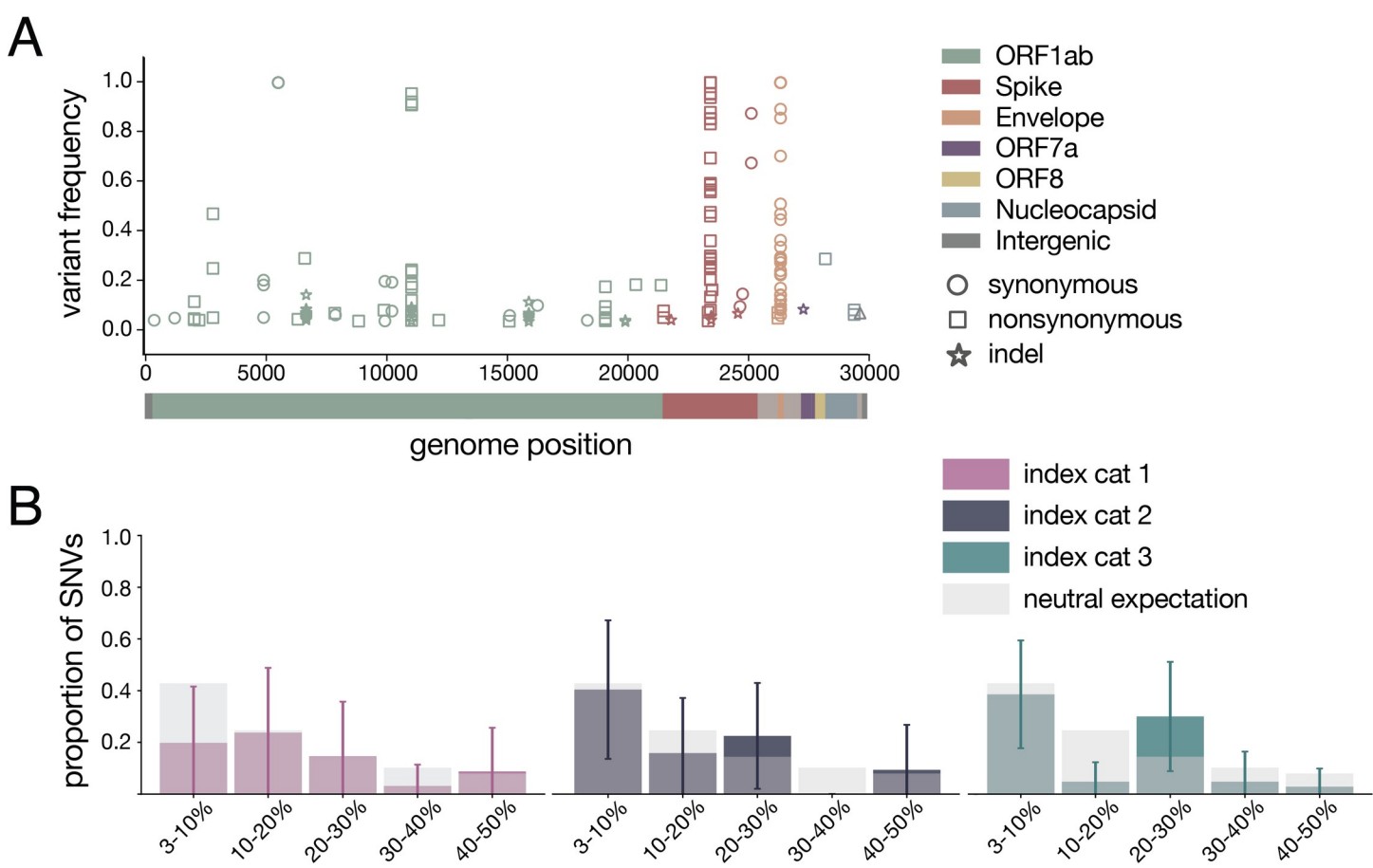

**Fig 2. Within-host diversity of SARS-CoV-2 viruses in domestic cats.** A) Plot representing all variants (iSNVs and indels) detected in any cat at any timepoint. Variant frequencies are plotted by genome location and are colored by gene. Circles represent synonymous iSNVs, squares represent nonsynonymous iSNVs, and stars represent indels. B) iSNV frequency spectrums with error bars showing standard deviation for index cats plotted against a "neutral model" (light gray bars) which assumes a constant population size and the absence of selection.

index cats 2 and 3 in ORF8 (p = 0.03, p = 0.04, unpaired t-test). πS and πN are not significantly different in at least one index cat in ORF3a, envelope, and nucleocapsid. There was not enough genetic variation to measure nucleotide diversity in the remaining four genes (**S1 Table**). Taken together, these results suggest longitudinal genetic variation within feline hosts is principally shaped by genetic drift with purifying selection acting on individual genes, particularly ORF1ab and Spike.

## Longitudinal sampling reveals few consensus-level changes within hosts

The consensus sequences recovered from all three index cats on the first day post-inoculation was identical to the inoculum or "stock" virus. This consensus sequence remained largely unchanged throughout infection in all index cats with the notable exception of two variants: H655Y in Spike (nucleotide site 23,525) and a synonymous change at amino acid position 67 in envelope (nucleotide site 26,445; S67S), which arose rapidly in all 3 index cats and rose to consensus levels ($\geq$50% frequency) at various timepoints throughout infection in all index cats. Neither of these iSNVs were detected above 3% frequency in the inoculum, but when we mined all sequencing reads, S H655Y and E S67S could be detected at 0.85% and 0.34%,

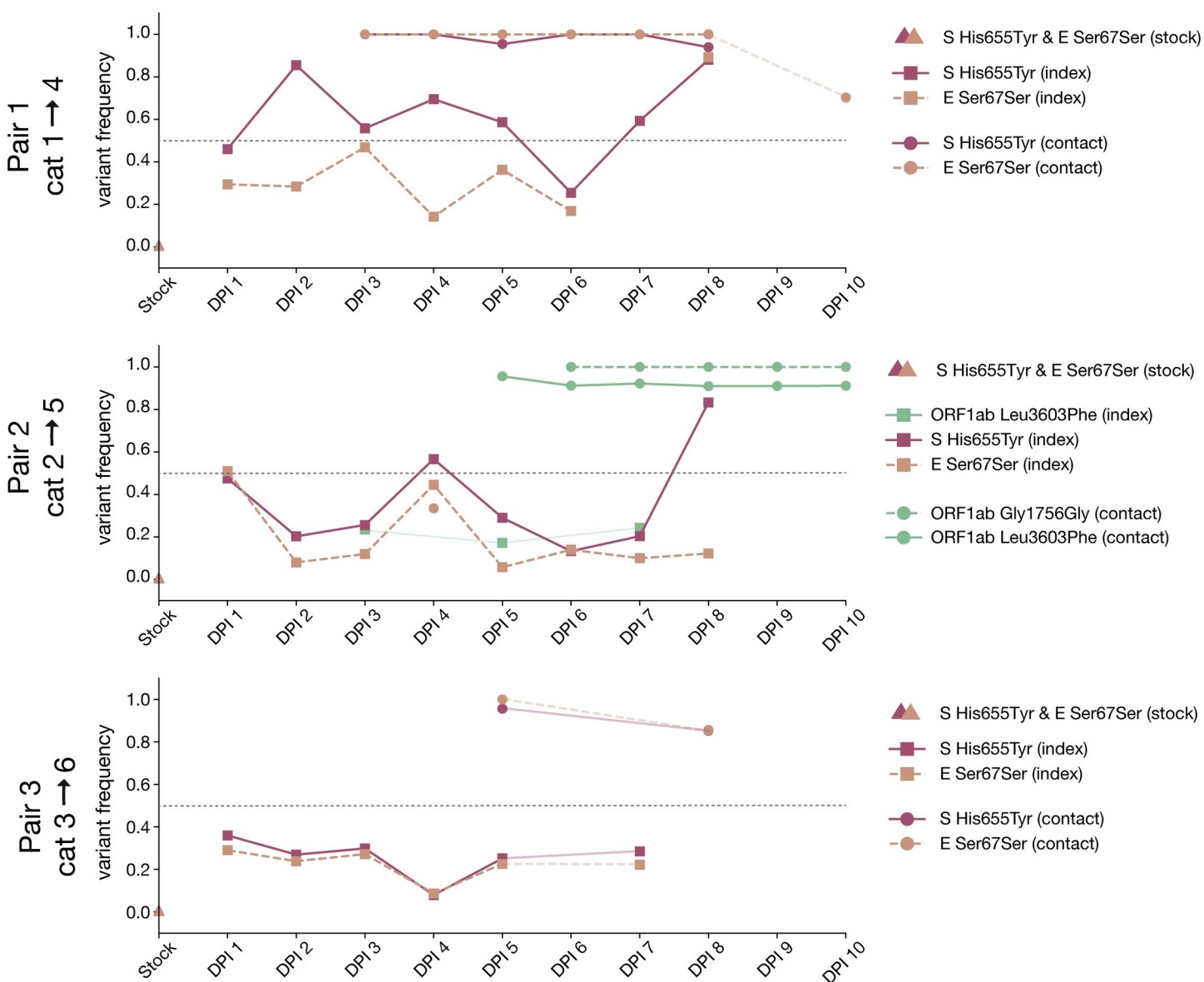

**Fig 3. Frequency of iSNVs over time in each index and contact cat.** The frequency of iSNVs discussed in the results over time in all six cats are shown. All iSNVs over time are shown in **S2 Fig** and all indels over time are shown in **S3 Fig**. Each variant is colored by gene location. Nonsynonymous variants are plotted with solid lines and synonymous variants are plotted with dashed lines. Variants detected in index cats are denoted with squares and variants detected in contact cats are denoted with circles. Timepoints with viral loads too low to yield high quality sequences are shown by the gaps in data, but iSNVs are connected across these gaps using light lines for readability (i.e. cat 1 day 9). The dotted line at 50% frequency represents the consensus threshold.

respectively. S H655Y was the consensus sequence on days 2–5 and days 7–8 in index cat 1, as well as on days 4 and 8 in index cat 2, and remained detectable above our 3% variant threshold throughout infection (**Fig 3**). Similarly, envelope S67S (E S67S) was the consensus sequence on day 8 in index cat 1 and day 1 in index cat 2. S H655Y and E S67S were detectable on days 1–7 in cat 3 but stayed below consensus level.

Interestingly, S H655Y and E S67S became fixed together following transmission in two transmission pairs (contact cats 4 and 6) and were lost together during transmission to contact animal 5. In cat 5, however, two different variants in ORF1ab, G1756G and L3606F, became

fixed after transmission. ORF1ab G1756G was not detected above 3% and L3606F was found at 17.2% in the day 5 sample from the index cat 2 (the cat transmitting to cat 5); it was not found in the inoculum at any detectable frequency. The categorical loss or fixation of these variants immediately following transmission, and in particular the fixation following transmission of a variant that was undetectable before, are highly suggestive of a narrow bottleneck [40].

In addition, a synonymous variant in an alanine codon at amino acid position 1,222 in Spike (nucleotide site 25,174) was found at >50% frequencies on days 4 and 8 in index cat 3, but was not detected above 3% on any other days. All iSNVs over time are shown in **S2 Fig** and all indels over time are shown in **S3 Fig**. These within-host analyses show that genetic drift appears to play a prominent role in shaping low-frequency genetic variation within hosts.

### SARS-CoV-2 transmission in domestic cats is defined by a narrow transmission bottleneck

To estimate the size of SARS-CoV-2 transmission bottlenecks, we investigated the amount of genetic diversity lost following transmission in cats. We observed a reduction in the cumulative number of variants detected in each contact cat compared to its index: 7 fewer variants in cat 4 (n = 9) compared to cat 1 (n = 16), 9 fewer in cat 5 (n = 10) than cat 2 (n = 19), and 10 fewer in cat 6 (n = 16) than cat 3 (n = 6). Likewise, the frequency distribution of variants in all three contact cats following transmission differed from the distribution of variants in all three index cats prior to transmission (p-value = 0.052, Mann Whitney U test). Following transmission, variant frequencies became more bimodally distributed than those observed in index cats, i.e., in contacts, most variants were either very low-frequency or fixed (**S2 Fig**).

To quantitatively investigate the stringency of each transmission event, we compared the genetic composition of viral populations immediately before and after viral transmission. We chose to use the first timepoint when infectious virus was recovered in the contact cat coupled with the timepoint immediately preceding this day in the index cat, as has been done previously [17]. We used days 2 (index) and 3 (contact) in pair 1, days 5 and 6 in pair 2, and days 4 and 5 in pair 3 (these sampling days are outlined in red in **Fig 1**). We applied the beta-binomial sampling method developed by Sobel-Leonard et al. to compare the shared set of variants (≥3%, ≤97%) in the pre/post-transmission timepoints for each pair [21]. Maximum-likelihood estimates determined that a mean effective bottleneck size of 5 (99% CI: 1–10), 3 (99% CI: 1–7), and 2 (99% CI: 1–3) best described each of the three cat transmission events evaluated here (**Fig 4**). This is in line with previous estimates for other respiratory viruses, including airborne transmission of seasonal influenza viruses in humans [40]. It is important to note, however, that the cat transmission pairs evaluated here shared physical enclosure spaces so the route of transmission could be airborne, direct contact, fomite, or a combination of these. Additionally, it has been shown that the route of influenza transmission can directly impact the size of the transmission bottleneck; for example, in one study airborne transmission of influenza viruses resulted in a narrow bottleneck, whereas contact transmission resulted in a wider bottleneck [16].

## Discussion

At the time of writing, the vast majority of humans remain immunologically naive to SARS-CoV-2. Whether through ongoing human adaptation, spill-back events from other animal intermediates, or with the distribution of vaccines and therapeutics, the landscape of SARS-CoV-2 variation is certain to change. Understanding the forces that shape genetic diversity of SARS-CoV-2 viruses within hosts will aid in forecasting the pace of genetic change as the virus faces shifting population-level immunity. Additionally, this baseline allows researchers to

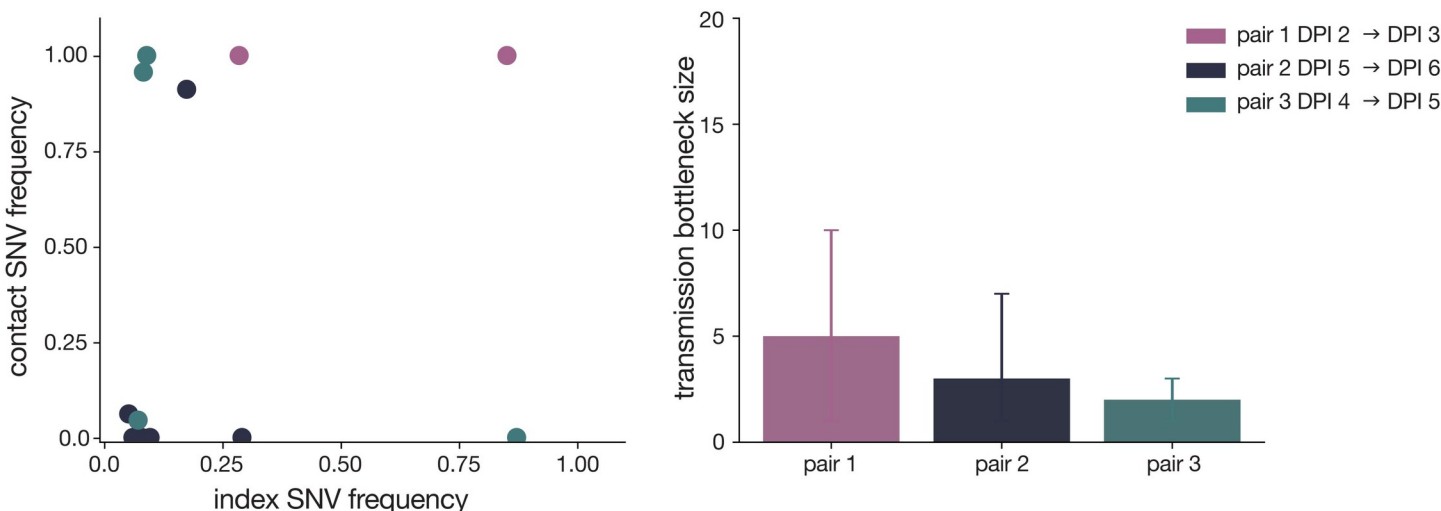

**Fig 4. SARS-CoV-2 transmission is defined by a narrow bottleneck.** Variant frequencies in the index cats (x-axis) compared with frequencies of the same variants in the corresponding contact cats (y-axis) that were used in the beta-binomial estimate are shown on the left. Estimates of SARS-CoV-2 transmission bottleneck with 99% confidence intervals shown on the right.

more easily identify a shift in the forces shaping within- and between-host diversity; for example, identification of signatures of positive selection might highlight rapidly-adapting, and therefore higher-risk, viruses.

Using domestic cats as a model system, we show stochastic processes like narrow transmission bottlenecks and genetic drift are major forces shaping SARS-CoV-2 genetic diversity within and between mammalian hosts. These stochastic forces typically act to constrain the overall pace of RNA virus evolution [12]. Despite this, we observe the rapid outgrowth of S H655Y in all three index cats, suggesting that this site may be under positive selection in this system. This variant achieved rapid fixation following transmission in two of three transmission pairs.

Our finding of narrow transmission bottlenecks is at odds with some recent studies in humans, which have estimated wide and variable SARS-CoV-2 transmission bottlenecks [25–28], but it is in line with other estimates suggesting that few SARS-CoV-2 viruses are transmitted between humans [25]. These discordant estimates are likely due to a combination of factors, including variable routes of transmission, uncertain sources of infection, difficulty collecting samples which closely bookend the transmission event, and inaccurate variant calls [25–28]. Human studies have commonly identified transmission pairs using intrahousehold infections diagnosed within a defined timeframe. A major weakness with this approach is the possibility that some of these cohabiting individuals will share an alternative source of exposure. Furthermore, without fine-scale epidemiological and clinical metadata, pinpointing the time of likely transmission is challenging, so even samples collected before and after a real transmission event may be several days removed from the time of transmission. Here we were able to circumvent many of these challenges by taking advantage of domestic cats experimentally infected with SARS-CoV-2 arranged in defined transmission pairs with clinical monitoring and daily sample collection, making for a useful model system.

The size of the transmission bottleneck may have additional implications for individual infections. The total number of founding virions, or the inoculum dose, has been posited to play a role in coronavirus disease 2019 (COVID-19) clinical severity and outcomes [41,42]. The transmission bottleneck can be parsed into two interdependent components: the

population bottleneck, or the number of virus particles that found infection (similar to inoculation dose); and the genetic bottleneck, or the amount of viral diversity lost during transmission. For example, an infection founded by 1,000 genetically identical viruses would be categorized as resulting from a narrow genetic bottleneck (a single genotype initiates the infection) and a relatively large population bottleneck. The beta-binomial method used here measures the population bottleneck [21]. Our data are consistent with a narrow population bottleneck and therefore a low inoculum dose in these cats. The extent to which feline hosts experience symptoms when infected with SARS-CoV-2 is unclear, but the cats involved in this study remained afebrile throughout the study, did not lose body weight, and experienced no respiratory signs. Viral genetic diversity has been linked to pathogenesis and clinical outcomes in the context of other viruses (e.g., influenza A virus, polio, and respiratory syncytial virus) and because narrow transmission bottlenecks often reduce viral genetic diversity, bottlenecks may play an essential role in the outcome of individual infections in this way as well [43–47]. The relationship between SARS-CoV-2 viral genetic diversity and COVID-19 clinical severity remains unclear. Some have proposed a direct relationship between particular viral lineages and COVID-19 severity [48], while others postulate that host factors, like age and comorbidities such as hypertension, diabetes, and preexisting respiratory system disease, are more likely to explain variable clinical outcomes [49].

Although within-host diversity was limited in the cats evaluated here, we identify two notable variants. S H655Y and E S67S were found at 0.85% and 0.34% in the stock, but were preferentially amplified in all three index cats and were detectable at intermediate frequencies at the first-day post-inoculation. Interestingly, S H655Y is not found in any of the 18 full-genome domestic cat, tiger, and lion SARS-CoV-2 sequences available on GISAID (**S4 Fig**). S H655Y has, however, been reported in a variety of other settings, including transmission studies in a hamster model, SARS-CoV-2 tissue culture experiments [50–53], and in a stock virus passaged on Vero E6 cells [BioProject PRJNA645906, experiment numbers SRX9287152 (p1), SRX9287151 (p2), SRX9287154 (p3a); BioProject PRJNA627977]. S H655Y additionally persisted in vivo in rhesus macaques challenged with one of these stock viruses [BioProject PRJNA645906, experiment number SRX9287155]. As of 28 December, 2020, S H655Y has been detected in 1,070 human SARS-CoV-2 viruses across 18 different countries in sequences deposited in GISAID. The majority of these sequences come from the United Kingdom (n = 886) (**S5B and S5C Fig**). It is important to note, however, that sampling of SARS-CoV-2 sequences is heavily biased and sequences from the COVID-19 Genomics UK consortium (COG-UK) are currently overrepresented in GISAID. At the time of writing, S H655Y was the 16[th] most common variant detected in Spike among publicly-available SARS-CoV-2 sequences [54]. Sequences containing S H655Y variant are found in two distinct European clusters, EU1 and EU2, suggesting it has arisen more than once (**S5A Fig**).

Relatively little is known about the phenotypic impact of S H655Y in cats, humans, and other host species. Amino acid residue 655 is located near the polybasic cleavage site, residing between the receptor binding domain (RBD) and the fusion peptide, and therefore has been hypothesized to play a role in regulating Spike glycoprotein fusion efficiency (**S12 Fig**) [50,51,55]. In spite of its location outside of the RBD, S H655Y has been shown to arise on the background of a vesicular stomatitis virus (VSV) pseudotyped virus expressing various SARS-CoV-2 spike variants and confer escape from multiple monoclonal human antibodies in cell culture [50]. It is unlikely S H655Y represents a site of antibody escape in these cats because they were specific pathogen-free and had undetectable IgG antibody titers against SARS-CoV-2 Spike and Nucleocapsid proteins on the day of infection [33]. We did not do any experiments to elucidate the functional impact of this variant, but we speculate S H655Y could have

improved Spike fusion efficiency and therefore host cell entry in cats. It is possible S H655Y offers a similar advantage in human hosts and/or confers escape from some antibodies.

E S67S has not been documented elsewhere. Based on iSNV frequencies, S H655Y and E S67S appear to be in linkage with each other (see mirrored iSNV frequencies in cat 2 and cat 5 in Fig 3 in particular), however with short sequence reads and sequencing approaches relying on amplicon PCR, we cannot rigorously assess the extent of linkage disequilibrium between these variants. It may be that S H655Y arose on the genetic background of an existing S67S variant in envelope. If S H655Y facilitates viral entry or replication in cats, viruses with this variant in linkage with E S67S might have been positively selected in all index cats.

Our data alone cannot resolve the precise mechanisms by which SARS-CoV-2 diversity is reduced during transmission, but the trajectories of S H655Y and E S67S raise some interesting possibilities. Although our sample size is small, the outgrowth of S H655Y with E S67S in all index cats, and the fixation of these variants in 2 of 3 contact cats, suggest that selection for one or both of these variants could have played a role in shaping genetic diversity recovered from contact cats. Viruses bearing these mutations could be preferentially amplified prior to, during, and/or after transmission.

If the transmission bottleneck is narrow and random, a variant's likelihood of being transmitted is equal to its frequency in the viral population at the time of transmission. If selection acts primarily within index hosts prior to transmission, S H655Y could have achieved a high enough frequency to be randomly drawn at the time of transmission. In this case, even a random, narrow transmission bottleneck could have facilitated the rapid fixation of a putatively beneficial variant. Next, suppose that viruses bearing S H655Y are shed more efficiently from index animals. In this case, evidence of selection in index animals would be limited and we would observe a small founding population in contact hosts where the beneficial variant is dominant. Alternatively, suppose viruses bearing S H655Y preferentially found infection in the recipient. In this case where selection is acting primarily in the contact host, transmission may involve transfer of a larger virus population after which beneficial variants may rapidly be swept to fixation. These scenarios are not mutually exclusive and it is possible for selection to act in concert before, during, and after transmission. In any of these scenarios, we would observe a low-diversity virus population in contact animals in which the putatively beneficial variants had been enriched. Notably, S H655Y and E S67S are absent from contact cat 5 (pair 2), despite being detectable and even reaching consensus levels in the associated index animal. While these variants are lost during transmission in this pair, a variant in ORF1ab (Gly1756Gly), which was undetectable in index cat 2, became fixed in contact cat 5 following transmission. The dramatic shifts in iSNV frequency we observe in all 3 pairs are characteristic of a narrow transmission bottleneck [12]. Because narrow transmission bottlenecks can result in the loss of even beneficial variants, the fact that S H655Y and E S67S failed to be transmitted in pair 2 does not exclude the possibility that these variants enhance viral fitness. Altogether our data therefore support the conclusion that SARS-CoV-2 transmission bottlenecks are narrow in this system, and may sometimes involve selection.

SARS-CoV-2 viruses can replicate and be shed via the respiratory tract. Differences in cell types, receptor distribution, temperature and humidity along the length of the respiratory tract may favor the emergence of different viral variants. If viral populations vary genetically across anatomic location, virus collected from different parts of the respiratory tract could result in different bottleneck size estimates. In this study, we had access to nasal swabs and therefore were only able to evaluate genetic diversity arising in the upper respiratory tract. Others have previously documented foci of influenza virus in the lower respiratory tract appear to be independent from upper respiratory tract infections [56,57]. Current insights into potential

differences in the genetic composition, structure, and evolution in the upper vs. lower respiratory tract remain incomplete for both influenza viruses and SARS-CoV-2.

Large SARS-CoV-2 outbreaks in mink have been reported recently, some with "concerning" mutations that may evade human humoral immunity [58]. These mink outbreaks have resulted in the Danish authorities' decision to cull 17 million mink as a safeguard against spillback transmission into humans [58]. Similarly, the emergence of the B.1.1.7 SARS-CoV-2 lineage has brought to light the importance of detecting and characterizing novel variants which might confer increased transmissibility, infectiousness, clinical severity, or other phenotypic change. The precise origins of the defining B.1.1.7 variants are unknown. It has been speculated that it may have arisen from a chronically infected patient or through sub-curative doses of convalescent plasma [59]. While S H655Y has not been found in mink and is not one of the defining B.1.1.7 mutations, another one of the defining B.1.1.7 mutations, Spike N501Y, has emerged independently in mouse models [60]. This suggests that mammalian models can facilitate the detection of novel mutations and signatures of positive selection, which might highlight adaptive mutations. We observe one variant that arises early and is transmitted onward in cats, a potential reservoir and model species. Little has been specifically documented about this variant, but it was very interesting to note it confers escape from various human monoclonal antibodies and has been detected in more than 1,000 human viruses [50,61]. Our study and the mink example show that species- and context-specific adaptations are likely to continue to emerge as SARS-CoV-2 explores new hosts. Further investigation and ongoing surveillance for such variants is warranted. It is also important to prevent the reintroduction of such newly formed variants, of which we do not know the potential phenotypic impacts, by limiting the spread and evolution of SARS-CoV-2 in non-human reservoir species.

As SARS-CoV-2 continues to spread globally, we must have models in place to recapitulate key evolutionary factors influencing SARS-CoV-2 transmission. With the imminent release of SARS-CoV-2 vaccines and therapeutics and increasing prevalence of natural exposure-related immunity, these models can help us forecast the future of SARS-CoV-2 variation and population-level genetic changes. Continued efforts to sequence SARS-CoV-2 across a wide variety of hosts, transmission routes, and spatiotemporal scales will be necessary to determine the evolutionary and epidemiological forces responsible for shaping within-host genetic diversity into global viral variation.

## Methods

### Ethics statement

No animal experiments were specifically performed for this study. We used residual nasal swabs collected from domestic cats as part of a previously published study [33]. Animal studies were approved prior to the start of the study by the Institutional Animal Care and Use Committee and performed in accordance with the Animal Care and Use Committee guidelines at the University of Wisconsin-Madison.

### Domestic cat experiments

No animal experiments were specifically performed for this study. We used residual nasal swabs collected from domestic cats as part of a previously published study [33]. Animals used in this study were specific-pathogen-free animals from a research colony maintained at the University of Wisconsin-Madison and were negative for feline coronavirus. As previously described by Halfmann et al, domestic cats were housed in 0.56 m x 0.81 m x 1.07 m cages in a laboratory with 65% humidity at 23˚C, and with at least 15.2 air exchanges per hour. Weight and body temperature (through implanted transponders) were measured daily (days 1–14).

Under ketamine and dexdomitor anesthesia, three cats were inoculated with 5.2 x 105 plaque-forming units (PFU of SARS-CoV-2 given by a combination of inoculation routes for every animal (nasal [100 µl per nare], tracheal [500 µl], oral [500 µl], and ocular [50 µl per eye]). To reverse the effects of the anesthesia, antisedan was administered to the animals after completion of the inoculation. Nasal swabs were collected daily during the study (days 1–10).

### Nucleic acid extraction

For each sample, approximately 140 µL of viral transport medium was passed through a 0.22 µm filter (Dot Scientific, Burton, MI, USA). Total nucleic acid was extracted using the Qiagen QIAamp Viral RNA Mini Kit (Qiagen, Hilden, Germany), substituting carrier RNA with linear polyacrylamide (Invitrogen, Carlsbad, CA, USA) and eluting in 30 µL of nuclease-free $H_2O$.

### Complementary DNA (cDNA) generation

Complementary DNA (cDNA) was synthesized using a modified ARTIC Network approach [34,35]. Briefly, RNA was reverse transcribed with SuperScript IV Reverse Transcriptase (Invitrogen, Carlsbad, CA, USA) using random hexamers and dNTPs. Reaction conditions were as follows: 1 µL of random hexamers and 1 µL of dNTPs were added to 11 µL of sample RNA, heated to 65˚C for 5 minutes, then cooled to 4˚C for 1 minute. Then 7 µL of a master mix (4 µL 5x RT buffer, 1 µL 0.1M DTT, 1µL RNaseOUT RNase Inhibitor, and 1 µL SSIV RT) was added and incubated at 42˚C for 10 minutes, 70˚C for 10 minutes, and then 4˚C for 1 minute.

### Multiplex PCR for SARS-CoV-2 genomes

A SARS-CoV-2-specific multiplex PCR for Nanopore sequencing was performed, similar to amplicon-based approaches as previously described [34,35]. In short, primers for 96 overlapping amplicons spanning the entire genome with amplicon lengths of 500bp and overlapping by 75 to 100bp between the different amplicons were used to generate cDNA. Primers used in this manuscript were designed by ARTIC Network and are shown in **S3 Table**. cDNA (2.5 µL) was amplified in two multiplexed PCR reactions using Q5 Hot-Start DNA High-fidelity Polymerase (New England Biolabs, Ipswich, MA, USA) using the following cycling conditions; 98˚C for 30 seconds, followed by 25 cycles of 98˚C for 15 seconds and 65˚C for 5 minutes, followed by an indefinite hold at 4˚C [34,35]. Following amplification, samples were pooled together before TrueSeq Illumina library prep.

### TrueSeq Illumina library prep and sequencing

Amplified cDNA was purified using a 1:1 concentration of AMPure XP beads (Beckman Coulter, Brea, CA, USA) and eluted in 30 µL of water. PCR products were quantified using Qubit dsDNA high-sensitivity kit (Invitrogen, USA) and were diluted to a final concentration of 2.5 ng/µl (150 ng in 50 µl volume). Each sample was then made compatible with deep sequencing using the TruSeq sample preparation kit (Illumina, USA). Specifically, each sample was enzymatically end repaired. Samples were purified using two consecutive AMPure bead cleanups (0.6x and 0.8x) and were quantified once more using Qubit dsDNA high-sensitivity kit (Invitrogen, USA). A non-templated nucleotide was attached to the 3′ ends of each sample, followed by adaptor ligation. Samples were again purified using an AMPure bead cleanup (1x) and eluted in 25 µL of resuspension buffer. Lastly, samples were amplified using 8 PCR cycles, cleaned with a 1:1 bead clean-up, and eluted in 30 µL of RSB. The average sample fragment length and purity was determined using the Agilent High Sensitivity DNA kit and the Agilent

2100 Bioanalyzer (Agilent, Santa Clara, CA). After passing quality control measures, samples were pooled equimolarly to a final concentration of 4 nM, and 5 μl of each 4 nM pool was denatured in 5 μl of 0.2 N NaOH for 5 min. Sequencing pools were denatured to a final concentration of 10 pM with a PhiX-derived control library accounting for 1% of total DNA and was loaded onto a 500-cycle v2 flow cell. Average quality metrics were recorded, reads were demultiplexed, and FASTQ files were generated on Illumina's BaseSpace platform.

## Processing of the raw sequence data, mapping, and variant calling

Raw FASTQ files were analyzed using a workflow called "SARSquencer". Briefly, reads are paired and merged using BBMerge (https://jgi.doe.gov/data-and-tools/bbtools/bb-tools-user-guide/bbmerge-guide/) and mapped to the reference (MW219695.1) using BBMap (https://jgi.doe.gov/data-and-tools/bbtools/bb-tools-user-guide/bbmap-guide/). Mapped reads were imported into Geneious (https://www.geneious.com/) for visual inspection. Read coverage for index cat samples is plotted in **S6 Fig** and for contact samples in **S7 Fig**. Variants were called using callvariants.sh (contained within BBMap) and annotated using SnpEff (https://pcingola.github.io/SnpEff/). The complete "SARSquencer" pipeline is available in the GitHub accompanying this manuscript in 'code/SARSquencer' as well as in a separate GitHub repository–https://github.com/gagekmoreno/SARS_CoV_2_Zequencer. BBMap's output VCF files were cleaned using custom Python scripts, which can be found in the GitHub accompanying this manuscript (https://github.com/katarinabraun/SARSCoV2_transmission_in_domestic_cats) [60]. Variants were called at ≥0.01% in reads that were ≥100 bp in length and supported by a minimum of 10 reads. Only variants at ≥3% frequency in both technical replicates were used for downstream analysis. Variant concordance across technical replicates is plotted in **S8 Fig** for index cats and **S9 Fig** for contact cats. In addition, all variants occurring in ARTIC v3 primer-binding sites were discarded before proceeding with downstream analysis.

## Quantification of SARS-CoV-2 vRNA

Plaque forming unit analysis was performed on all nasal swabs as published in Halfmann et al. 2019 [33]. Viral load analysis was performed on all of the nasal swab samples described above after they arrived in our laboratory. RNA was isolated using the Viral Total Nucleic Acid kit for the Maxwell RSC instrument (Promega, Madison, WI) following the manufacturer's instructions. Viral load quantification was performed using a sensitive qRT-PCR assay developed by the CDC to detect SARS-CoV-2 (specifically the N1 assay) and commercially available from IDT (Coralville, IA). The assay was run on a LightCycler 96 or LC480 instrument (Roche, Indianapolis, IN) using the Taqman Fast Virus 1-stepMaster Mix enzyme (Thermo Fisher, Waltham, MA). The limit of detection of this assay is estimated to be 200 genome equivalents/ml saliva or swab fluid. To determine the viral load, samples were interpolated onto a standard curve consisting of serial 10-fold dilutions of in vitro transcribed SARS-CoV-2 N gene RNA.

## Pairwise nucleotide diversity calculations

Nucleotide diversity was calculated using π summary statistics (**S2 Table**). π quantifies the average number of pairwise differences per nucleotide site among a set of sequences and was calculated per gene using SNPGenie (https://github.com/chasewnelson/SNPgenie) [62]. SNPGenie adapts the Nei and Gojobori method of estimating nucleotide diversity (π), and its synonymous (πS) and nonsynonymous (πN) partitions from next-generation sequencing data [63]. When πN = πS, this indicates neutral evolution or genetic drift, with neither strong purifying nor positive selection playing a large role in the evolution of the viral population. πN <

πS indicates purifying selection is acting to remove deleterious mutations, and πN > πS shows positive or diversifying selection acting on nonsynonymous variation [64]. We tested the null hypothesis that πN = πS within each gene using an unpaired t-test (**S1 Table**). The code to replicate these results can be found in the 'diversity_estimates.ipynb' Jupyter Notebook in the 'code' directory of the GitHub repository [65].

## SNP Frequency Spectrum calculations

To generate SNP Frequency Spectrums (SFS), we binned all variants detected across time-points within each index cat into six bins– 3–10%, 10–20%, 20–30%, 30–40%, 40–50%, 50–60%. We plotted the counts of variants falling into each frequency bin using Matplotlib 3.3.2 (https://matplotlib.org). We used code written by Dr. Louise Moncla to generate the distribution of SNPs for a given population assuming no selection or change in population size, which is expected to follow a 1/x distribution [37]. The code to replicate this can be found in the GitHub accompanying this manuscript, specifically in the 'code/SFS.ipynb' Jupyter Notebook. This model predicts 42.8% of variants will fall within the 3–10% frequency range, 24.6% will fall within the 10–20% frequency range, 14.4% of variants will fall within the 20–30% frequency range, 10.2% of variants will fall within the 30–40% frequency range, and 7.9% of variants will fall within the 40–50% frequency range. We used a Mann-Whitney U test to test the null hypothesis that the distribution of variant frequencies for each index cat was equal to the neutral distribution. The code to replicate these results can be found in the 'SFS.ipynb' Jupyter Notebook in the 'code' directory of the GitHub repository [65].

## Focal Nextstrain build of S H655Y sequences

The focal H655Y build (**S5 Fig**) was prepared as described in Hodcroft et al. (2020), with different mutations targeted for the S:655 mutation [66]. Briefly: sequences with a mutation at nucleotide position 23525 (corresponding to a change at the 655 position in the spike glycoprotein) were selected from all available sequences on GISAID as of 29th December 2020. These sequences were included as the 'focal' set for a Nextstrain phylogenetic analysis, to which 'context' sequences were added, with the most genetically similar sequences given priority.

## Code and data availability

Code to replicate analyses and re-create most figures is available at https://github.com/katarinabraun/SARSCoV2_transmission_in_domestic_cats [65]. **Fig 1** was created by hand in Adobe Illustrator and **S6 and S7 Figs** were created using samtools command line tools, were visualized in JMP Pro 15 (https://www.jmp.com/en_in/software/new-release/new-in-jmp-and-jmp-pro.html), and were then edited for readability in Adobe Illustrator. Code to process sequencing data is available at https://github.com/gagekmoreno/SARS_CoV_2_Zequencer and dependencies are available through Docker [67]. Results were visualized using Matplotlib 3.3.2 (https://matplotlib.org), Seaborn v0.10.0 (https://github.com/mwaskom/seaborn), and Baltic v0.1.0 (https://github.com/evogytis/baltic).

## Supporting information

**S1 Fig. Viral loads and viral titers over time.** A) Viral RNA burden over time for each cat. Index cats are represented by a solid line and contact cats are represented by a dashed line. Transmission pairs are denoted by color. The grey, horizontal dotted line represents when less than ~100 copies/μL are input into the reverse transcription reaction. B) Infectious viral titer

over time. Index cats are represented by a solid line and contact cats are represented by a dashed line. Transmission pairs are denoted by color.
(TIF)

**S2 Fig. Longitudinal frequency of iSNVs detected in all cats and at all timepoints.** Each variant is colored based on gene location. Nonsynonymous variants are plotted with solid lines and synonymous variants are plotted with dashed lines. Days with viral loads too low to yield high quality sequences are shown by the gaps in data (i.e. cat 3 day 6 and cat 4 day 9).
(TIF)

**S3 Fig. Longitudinal frequency of indels detected in all cats and at all timepoints.** Each indel is colored based on gene location. Days with viral loads too low to yield high quality sequences are shown by the gaps in data (i.e. cat 3 day 6 and cat 4 day 9). Note the y-axis range is 0–12%, not 0–100%, to facilitate readability.
(TIF)

**S4 Fig. Sequence alignment of all tiger, lion, and domestic cat sequences available in GISAID as of December 2020.** Sequences were aligned against MW219695.1, the inoculum virus used in these experiments. Consensus-level differences are highlighted with a blue vertical marker. Indels are noted with a horizontal vertical marker. The spike open reading frame is annotated with a green marker and site amino acid 655 in Spike is highlighted with the orange box. None of these sequences contain a consensus mutation at residue 655 in Spike.
(TIF)

**S5 Fig. Geographic dispersion of Spike H655Y variant.** A) A time-resolved phylogeny focused on viruses that contain Spike H655Y. Viruses that contain histidine (H) at Spike 655 are colored in teal. Viruses with tyrosine (Y) at Spike 655 are colored in yellow. B) Counts of SARS-CoV-2 viruses that contain Spike H655Y, broken down by country. C) Map highlighting the number viruses from each country. The size of the circle represents the number of sequences from the appropriate country contained in the phylogeny.
(TIF)

**S6 Fig. Read depth across the SARS-CoV-2 genome in index cats.** Each day is represented by a different color. Replicate A is shown in the left column and replicate B is shown in the right column.
(TIF)

**S7 Fig. Read depth across the SARS-CoV-2 genome in contact cats.** Each day is represented by a different color. Replicate A is shown in the left column and replicate B is shown in the right column.
(TIF)

**S8 Fig. Intersection variants found across technical replicates in index cats.** The frequency of each variant per replicate is shown here. The diagonal line represents the 1:1 intersection of replicate variants. The subplot to the right of each primary plot is a zoomed-in view of the low-frequency variants (3–15%). Each timepoint is denoted by a different color.
(TIF)

**S9 Fig. Intersection variants found across technical replicates in contact cats.** The frequency of each variant per replicate is shown here. The diagonal line represents the 1:1 intersection of replicate variants. The subplot to the right of each primary plot is a zoomed-in view of the low-frequency variants (3–15%). Each timepoint is denoted by a different color.
(TIF)

**S10 Fig. Longitudinal pairwise nonsynonymous nucleotide diversity divided by pairwise synonymous nucleotide diversity in index cats.** Line color denotes gene. The horizontal dotted gray line is plotted at $y = 1$ or when $\pi N \sim \pi S$.
(TIF)

**S11 Fig. Longitudinal pairwise nonsynonymous nucleotide diversity divided by pairwise synonymous nucleotide diversity in contact cats.** Line color denotes gene. The horizontal dotted gray line is plotted at $y = 1$ or when $\pi N \sim \pi S$.
(TIF)

**S12 Fig. SARS-CoV-2 spike glycoprotein crystal structure.** Spike H655Y is highlighted in blue. The table to the right of the crystal structure includes summary information regarding the impact of a histidine to tyrosine change on amino acid charge, volume, and aromaticity. * Qualitative definitions of radical amino acid replacements, based on three alternative residue groupings, see Hanada et al., 2006 [68]. The crystal structure and summary information were generated using GISAID's CoVserver mutation analysis tool.
(TIF)

**S1 Table. Nonsynonymous and synonymous nucleotide diversity estimates in index cats.**
(PDF)

**S2 Table. Genome-wide pairwise nucleotide diversity estimates in index and contact cats.**
(PDF)

**S3 Table. ARTIC v3 primer sequences.**
(PDF)

## Acknowledgments

We would like to acknowledge Genetic Services (https://ehr.primate.wisc.edu/project/WNPRC/WNPRC_Units/Research_Services/Genetics_Services/Public/begin.view?) who sequenced the challenge stock viruses discussed in the discussion section of this paper (BioProject: PRJNA627977). We thank Chelsea Crooks for her careful reading of and comments on this manuscript.

## Author Contributions

**Conceptualization:** Katarina M. Braun, Gage K. Moreno, Peter J. Halfmann, Masato Hatta, Shiho Chiba, Tadashi Maemura, Yoshihiro Kawaoka, Thomas C. Friedrich.

**Data curation:** Katarina M. Braun, Gage K. Moreno, Peter J. Halfmann, David A. Baker, Masato Hatta.

**Formal analysis:** Katarina M. Braun, Gage K. Moreno, Katia Koelle.

**Funding acquisition:** Yoshihiro Kawaoka, David H. O'Connor, Thomas C. Friedrich.

**Investigation:** Katarina M. Braun, Gage K. Moreno, Peter J. Halfmann, Emma B. Hodcroft, Emma C. Boehm, Andrea M. Weiler, David H. O'Connor, Thomas C. Friedrich.

**Methodology:** Katarina M. Braun, Gage K. Moreno, Katia Koelle.

**Project administration:** Katarina M. Braun, Gage K. Moreno, David H. O'Connor, Thomas C. Friedrich.

**Resources:** Peter J. Halfmann, Emma B. Hodcroft, Yoshihiro Kawaoka.

**Software:** Katarina M. Braun, Gage K. Moreno, David A. Baker, Amelia K. Haj.

**Supervision:** Yoshihiro Kawaoka, David H. O'Connor, Thomas C. Friedrich.

**Validation:** Katarina M. Braun, Gage K. Moreno, Peter J. Halfmann.

**Visualization:** Katarina M. Braun, Gage K. Moreno, Emma B. Hodcroft.

**Writing – original draft:** Katarina M. Braun, Gage K. Moreno, Emma C. Boehm, David H. O'Connor, Thomas C. Friedrich.

**Writing – review & editing:** Katarina M. Braun, Gage K. Moreno, Peter J. Halfmann, Emma B. Hodcroft, David A. Baker, Emma C. Boehm, Andrea M. Weiler, Amelia K. Haj, Masato Hatta, Shiho Chiba, Tadashi Maemura, Yoshihiro Kawaoka, Katia Koelle, David H. O'Connor, Thomas C. Friedrich.

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
