## [Decision Letter · Decision Letter 0]

28 Jan 2021

Dear Dr. Friedrich,

Thank you very much for submitting your manuscript "Transmission of SARS-CoV-2 in domestic cats imposes a narrow bottleneck" for consideration at PLOS Pathogens. As with all papers reviewed by the journal, your manuscript was reviewed by members of the editorial board and by several independent reviewers. The reviewers appreciated the attention to an important topic. Based on the reviews, we are likely to accept this manuscript for publication, providing that you modify the manuscript according to the review recommendations.

You'll find that both reviewers found the study interesting and important and they have made constructive comments on the manuscript. While the manuscript is focused on genetic changes in the SARS-CoV-2 genome associated with transmission bottlenecks, both reviewers commented on the importance of the H655Y mutation and made suggestions regarding experiments that could address the impact of H655Y on S protein function and antibody responses. While both of those are interesting and important experiments, I wont' require you to address them experimentally in a revised manuscript. All other reviewer comments should be addressed in a revised manuscript.

Sincerely,

Andrew Pekosz, Ph.D.

Section Editor

PLOS Pathogens

Andrew Pekosz

Section Editor

PLOS Pathogens

Kasturi Haldar

Editor-in-Chief

PLOS Pathogens

orcid.org/0000-0001-5065-158X

Michael Malim

Editor-in-Chief

PLOS Pathogens

orcid.org/0000-0002-7699-2064

You'll find that both reviewers found the study interesting and important and they have made constructive comments on the manuscript. While the manuscript is focused on genetic changes in the SARS-CoV-2 genome associated with transmission bottlenecks, both reviewers commented on the importance of the H655Y mutation and made suggestions regarding experiments that could address the impact of H655Y on S protein function and antibody responses. While both of those are interesting and important experiments, I wont' require you to address them experimentally in a revised manuscript. All other reviewer comments should be addressed in a revised manuscript.

Reviewer Comments (if any, and for reference):

Reviewer's Responses to Questions

**Part I - Summary**

Reviewer #1: Braun et al have examined the transmission bottleneck of SARS-CoV-2 in domestic co-housed cats. In a serious of well-defined analyses, the authors define the SNV in donor and recipient animals to quantify the bottleneck and extrapolate that 2-5 infectious viruses likely seeds subsequent infections. In 2 of the 3 pairs of cats, the authors describe a potential selective advantage of Spike mutation H665Y and Envelope protein mutation S67S. The H665Y mutation has emerged in the human population and it interesting that is would in cats as well. However, this mutation was not observed in any of the deposited feline SARS-CoV-2 sequences, which is surprising.

Overall the authors provide a clear explanation of the bottleneck analysis and the discussion describes the nuance associated with these types of analyses. The bottleneck size of close contact transmission is important to understand as the authors demonstrate may be driven by a very small dose of virus particles. We provide a few suggestions that could increase the clarity and enhance the biological insight of this manuscript, but overall this article is of broad interest and timely.

1. The authors reference the robust replication of SARS-CoV-2 in domestic cats in multiple anatomical sites (ref 1, Shi et al). Are the authors aware of whether the H665Y mutation emerged in all anatomical sites, or is there a preference for the upper respiratory tract (given the site of nasal wash). The prevalence in other organs may influence the amount of this variant released into environment and could alter some of the bottleneck calculations. If that data are not available – perhaps a mention of how variations of viral adaptation within anatomical sites, could influence bottleneck calculations in the discussion is warranted.

2. The emergence of H665Y variants in Spike has caused a great deal of concern given the escape form certain monoclonal antibodies. It would be interesting if the authors could use the cat sera from donors (where is this variant is low) to demonstrate neutralization of viruses carrying this mutation. Data such as this could help address whether polyclonal responses, particular during natural infection would still recognize this new variant. The corresponding study describing the cat transmission experiments (ref 33, Halfmann et al), only specifies ELISA antibody titers, but neutralizing titers – may be more informative. However, this aspect may be beyond the scope of this manuscript.

Reviewer #2: In the present study by Braun and colleagues titled “Transmission of SARS-CoV-2 in domestic cats imposes a narrow bottleneck”, SARS-CoV-2 genetic diversity is assessed within and between hosts using the COVID-19 cat transmission model. The authors determined how SARS-CoV-2 might evolve as it continues to spread to various susceptible animal species. They could show that narrow transmission bottle necks and genetic drift are the major forces shaping SARS-CoV-2 genetic diversity in cats. The results also suggest that a positive selective pressure may exist for the H655Y spike variant which was identified in all index cats; this spike variant is also described in other experimental animal models, tissue culture studies and in the human population. The authors conclude that the COVID-19 cat transmission model recapitulates key aspects of SARS-CoV-2 evolution in humans. In the cat model, within host genetic variation is predominantly influenced by genetic drift and purifying selection and transmission between cats is defined by a narrow transmission bottleneck involving only 2-5 viruses. Therefore, the COVID-19 cat model might serve as a model system to study SARS-CoV-2 diversity and evolution within and between hosts.

**Part II – Major Issues: Key Experiments Required for Acceptance**

Reviewer #1: (No Response)

Reviewer #2: The manuscript is well written and contributes to our understanding of SARS-CoV-2 diversity and evolution. The introduction provides sufficient background and the conclusions appear to be supported by the data. The results are presented and illustrated clearly by the figures and discussed appropriately. The analysis appears to be thorough and the methods and statistics used appear appropriate. The identification of the SARS-CoV-2 variant H655Y being positively selected in cats is interesting. Since this variant was previously shown escape from monoclonal antibodies and the substitution is close to the fusion peptide, the authors hypothesize that this site in the spike protein may be under positive selection in cats and being potentially beneficial for cat replication. In this context, it is important to know whether the cats used in the experiment were seronegative not only for SARS-CoV-2-specific antibodies but also for antibodies against feline coronaviruses. In addition, data using H655Y pseudoviruses or plaque-purified/rescued H655Y variant viruses should be presented to help answering questions regarding the functional impact (e.g., fusion efficiency, host entry; immune escape) of the H655Y substitution. The question still remains why only two out of three contact cats selected the H655Y variant.

**Part III – Minor Issues: Editorial and Data Presentation Modifications**

Reviewer #1: (No Response)

Reviewer #2: Lines 268-271: What about comorbidities? Please clarify.

Figure 3: Please add respective cat numbers to the Figure.

Figure 4. Define Nb.

Suppl. Figure 3: It is very difficult to read this figure. Please improve coloring, etc.

PLOS authors have the option to publish the peer review history of their article (what does this mean?). If published, this will include your full peer review and any attached files.

Reviewer #1: No

Reviewer #2: No
---

## [Editor Report · Decision Letter 1]

12 Feb 2021

Dear Dr. Friedrich,

We are pleased to inform you that your manuscript 'Transmission of SARS-CoV-2 in domestic cats imposes a narrow bottleneck' has been provisionally accepted for publication in PLOS Pathogens.

Best regards,

Andrew Pekosz, Ph.D.

Section Editor

PLOS Pathogens

Andrew Pekosz

Section Editor

PLOS Pathogens

Kasturi Haldar

Editor-in-Chief

PLOS Pathogens

orcid.org/0000-0001-5065-158X

Michael Malim

Editor-in-Chief

PLOS Pathogens

orcid.org/0000-0002-7699-2064
---

## [Editor Report · Acceptance letter]

22 Feb 2021

Dear Dr. Friedrich,

We are delighted to inform you that your manuscript, "Transmission of SARS-CoV-2 in domestic cats imposes a narrow bottleneck," has been formally accepted for publication in PLOS Pathogens.

Best regards,

Kasturi Haldar

Editor-in-Chief

PLOS Pathogens

orcid.org/0000-0001-5065-158X

Michael Malim

Editor-in-Chief

PLOS Pathogens

orcid.org/0000-0002-7699-2064